# Use of Monoclonal Antibodies in Immunocompromised Patients Hospitalized with Severe COVID-19: A Retrospective Multicenter Cohort

**DOI:** 10.3390/jcm12030864

**Published:** 2023-01-21

**Authors:** Jorge Calderón-Parra, Pablo Guisado-Vasco, Rocío Montejano-Sánchez, Vicente Estrada, Guillermo Cuevas-Tascón, José Aguareles, José Arribas, Marta Erro-Iribarren, Marina Calvo-Salvador, Ana Fernández-Cruz, Antonio Ramos-Martínez, Elena Muñez-Rubio

**Affiliations:** 1Infection Diseases Unit, Internal Medicine Department, University Hospital Puerta de Hierro, 28222 Majadahonda, Spain; 2Research Institute Puerta de Hierro-Segovia de Aranda (IDIPHISA), 28222 Majadahonda, Spain; 3Infectious Disease Department, University Hospital Quironsalud Madrid, 28223 Madrid, Spain; 4Infectious Diseases Unit, University Hospital La Paz, 28046 Madrid, Spain; 5Infectious Disease Department, University Hospital Clínico San Carlos, 28040 Madrid, Spain; 6Internal Medicine Department, University Hospital Infanta Leonor, 28031 Madrid, Spain; 7Centro de Investigación Biomédica en Red de Enfermedades Infecciosas (CIBERINFEC), 28029 Madrid, Spain; 8Pneumology Department, University Hospital Puerta de Hierro, 28222 Majadahonda, Spain; 9Pharmacology Department, University Hospital Puerta de Hierro, 28222 Majadahonda, Spain

**Keywords:** COVID-19, monoclonal antibodies, sotrovimab, severe COVID-19, immunocompromised hosts

## Abstract

Objective: We aim to describe the safety and efficacy of sotrovimab in severe cases of COVID-19 in immunocompromised hosts. Methods: We used a retrospective multicenter cohort including immunocompromised hospitalized patients with severe COVID-19 treated with sotrovimab between October 2021 and December 2021. Results: We included 32 patients. The main immunocompromising conditions were solid organ transplantation (46.9%) and hematological malignancy (37.5%). Seven patients (21.9%) had respiratory progression: 12.5% died and 9.4% required mechanical ventilation. Patients treated within the first 14 days of their symptoms had a lower progression rate: 12.0% vs. 57.1%, *p* = 0.029. No adverse event was attributed to sotrovimab. Conclusions: Sotrovimab was safe and may be effective in its use for immunocompromised patients with severe COVID-19. More studies are needed to confirm these preliminary data.

## 1. Introduction

Monoclonal antibodies (mAbs) against SARS-CoV-2 have reduced hospitalization or death in outpatients with mild to moderate COVID-19 [1,2]. The results of randomized trials in hospitalized patients are controversial [3,4,5]. In addition, clinical trials testing these drugs were conducted in unvaccinated populations, with a possible lower efficacy in vaccinated patients [6].

mAbs have also not been studied in immunocompromised hosts. Some trials excluded immunocompromised hosts [1], while others included less than 2% of patients with these conditions [2]. The vaccine efficacy in this population is lower [7], and many will fail to mount an immune response when infected with SARS-CoV2 [8], so they represent a special population with a high risk of respiratory progression and death [8,9,10]. Accordingly, it has been proposed that research of mAbs in patients with severe COVID-19 should focus on this subgroup of patients [11].

Sotrovimab represents one alternative for these patients [1]. It is an engineered human monoclonal antibody that neutralizes multiple coronaviruses by targeting a highly conserved epitope. Sotrovimab can have activity against mutant-rich variants, such as omicron, including BA.1 and BA.2 [12,13]. It has been designed to have Fc effector functions, which provide additional activity [14,15]. However, this drug has not been studied in severe COVID-19 case, nor in immunocompromised hosts.

We aim to describe the safety and efficacy of sotrovimab in severe cases of COVID-19 in immunocompromised hosts in a setting with a predominance of mutant-rich variants.

## 2. Materials and Methods

We conducted a retrospective multicenter uncontrolled cohort study in five hospitals. Participant centers were Puerta de Hierro, Quironsalud Madrid, La Paz, Clinico San Carlos, and Infanta Leonor University Hospitals, all of them located in Madrid, Spain.

We included patients with severe COVID-19 infections for whom sotrovimab had been administrated after their hospital admission. We included patients between October 2021 and December 2021.

Inclusion criteria were:Age 18 years or older.Severe COVID-19, defined as requiring hospital admission plus oxygen supplementation and/or bilateral pneumonia.Presence of any immunocompromising condition, including:
a.Solid organ malignancy with active treatment.b.Active hematological malignancy.c.Solid organ transplantation.d.Hematopoietic stem cell transplantation.e.Chronic corticoid treatment prior to COVID-19.f.Immunosuppressive treatment.g.HIV infection with lymphocyte depletion (<300 CD4 T cells/mL).h.Primary immunodeficiency excluding isolated IgA deficit.

These inclusion criteria were in line with the prioritization protocol by Spanish Drug Agency (AEMPS for its Spanish abbreviation) [16].

There were no exclusion criteria.

Data were retrospectively collected from electronic medical records and managed using REDCap electronic capture tools [17], with licenses provided to Research Institute Puerta de Hierro-Segovia de Aranda.

### 2.1. Definitions

Severe COVID-19 was defined according to World Health Organization (WHO) guidelines [18]. PaFi is defined as the relation between partial oxygen pression and inspirated oxygen fraction, and SaFi is defined as the relation between transcutaneous oxygen saturation and inspirated oxygen fraction. The WHO 7-point ordinary scale was used to define clinical status [18]. Respiratory progression was defined as the need for invasive mechanical ventilation, intensive care unit (ICU) admission, and/or death due to COVID-19-related pulmonary injury.

### 2.2. Primary and Secondary Endpoints

The primary efficacy outcomes were respiratory progression at day 29, all-cause in-hospital mortality, and COVID-19-related mortality. Secondary outcomes included time to discharge, time to oxygen supplementation withdrawal, and clinical status at 2, 7, 14, and 29 days, and the presence of adverse reactions related to sotrovimab.

### 2.3. Statistical Analyses

Qualitative variables are expressed as absolute value and percentages. Quantitative and ordinary variables are expressed as median and interquartile range (IQR). For inferential statistics, Chi square test was used to compare categorical variables (or Fisher exact test when necessary). Mann–Whitney’s U was used to compare quantitative variables. All statistical analyses were performed using SPSS software version 25.

## 3. Results

Thirty-two (32) patients who received a single dose of 500 mg of sotrovimab were included. The median time from symptoms onset to infusion was 9 days (IQR 5–15). The median follow-up time was 50 days (IQR 38–71).

### 3.1. Patients Characteristics

Table 1 summarizes the patients’ baseline characteristics. The median age was 67 years (IQR 59–76) and 43.8% (*n* = 14) were female. The main immunocompromising conditions were solid organ transplantation 46.9% (*n* = 15, nine lung, four kidney, two heart, and one liver) and active hematological malignancy 37.5% (*n* = 12). The majority of patients were fully vaccinated (93.8%, *n* = 30). Nevertheless, anti-spike antibodies were undetectable in 91.0% of patients with available serology (*n* = 20/22). All patients presented bilateral interstitial pneumonia and the low-flow nasal cannula oxygen requirement.

### 3.2. Primary and Secondary Efficacy Endpoints

Seven patients (21.9%) had respiratory progression: 12.5% (*n* = 4) died due to COVID-19 and 9.4% (*n* = 3) required ICU admission but survived. All-cause mortality was 15.6% (*n* = 5) (Table 1).

The median admission duration was eleven days (IQR 8–22) and the median oxygen supplementation duration was seven days (IQR 5–10). Figure 1 shows the clinical status during the follow-up.

A PaFi greater than 210 at infusion was associated with a lower rate of respiratory progression: 11.5% (3/26) versus 66.7% (4/6), *p* = 0.005 (Figure 2). Those receiving sotrovimab within the first 14 days from symptom onset had a lower progression risk: 12.0% (*n* = 3/25) vs. 57.1% (*n* = 4/7), *p* = 0.029 (Figure 2).

### 3.3. Safety

No sotrovimab-related complications or adverse events were noted. COVID-19-related complications were noted in 40.6% (*n* = 13) of patients. The most frequent were severe acute respiratory distress syndrome 28.2% (*n* = 9), bacterial pneumonia 18.8% (*n* = 6), and acute renal injury 18.8% (*n* = 6).

## 4. Discussion

To the best of our knowledge, this is the first report of the use of sotrovimab in immunocompromised patients who are hospitalized for severe cases of COVID-19. Our main finding is that sotrovimab administration appears to be safe for these patients and may have potential clinical benefits.

Only a few other studies have evaluated the efficacy of mAbs in non-immunocompromised patients with severe COVID-19, with conflicting results [1,3]. Recovery trials have shown a possible modest effect in a subgroup of seronegative patients [3], suggesting that mAbs could be beneficial for a subset of patients unable to develop an effective immune response, such as immunocompromised patients [6,7]. One recent trial has also found that mAbs could be clinically beneficial in seronegative patients, with no clear benefit for seropositive patients [19]. Nevertheless, these papers included only a small proportion of immunocompromised patients and their conclusions may not be valid for this population.

Immunocompromised patients are at higher risk of ICU admission and death due to COVID-19 [9,10,20]. Vaccine protection in this group is lower [7,21], with frequent development severe COVID-19 after full vaccination [22,23,24]. In-hospital mortality among these patients is above 30% [9,10,22]. In one recent cohort of fully vaccinated patients, immunocompromised patients had a respiratory progression rate of 36%, with mortality rates close to 30% [22]. Another cohort has shown mortality rates above 30% in vaccinated immunosuppressed patients infected with omicron [25,26]. In comparison, in our cohort, respiratory progression and mortality rates (21.9% and 15.6%, respectively) were numerically lower. Of note, our rate of ICU admission was lower than that of a recent cohort, including vaccinated patients with no serologic response (12.5% vs. 51.1%, and 15.6% vs. 21.3%) [23]. A recent multicenter cohort has shown the potential benefits of monoclonal antibodies for immunosuppressed patients at any stage, including severe and critical cases of COVID-19 [25]. Additionally, other antiviral agents have shown benefits on the post-acute COVID-19 syndrome [27], which may be a potential benefit of mAbs in this population. Therefore, although we could not directly compare our treated patients with a proper control group, our data suggest that neutralizing mAbs may be useful in immunocompromised patients with severe COVID-19. However, prospective and comparative studies are needed before we can extract robust conclusions of the efficacy of this treatment.

Additionally, we have found factors associated with a good clinical response after sotrovimab infusion. Sotrovimab-treated patients within the first 14 days of symptoms and with low-oxygen supplementation requirements had a lower respiratory progression. Our SaFi and PaFi thresholds are equivalent to the requirement of FiO_2_ equal or higher than 40% (low-flow nasal cannula at 5 L/min). Our results suggest that mAbs treatment for immunosuppressed patients hospitalized with severe COVID-19 could be focused on those with oxygen supplementation with a flow below 5 L/min by low-flow nasal cannula and symptoms fewer than 14 days from symptoms onset. Once again, prospective controlled studies are needed before reaching robust conclusions on which subgroups of patients may benefit the most.

On the other hand, cases of breakthrough COVID-19 are more frequent if infected by the new variants, such as omicron [28]. Accordingly, it can be speculated that only mAbs, with a retained activity to these variants, could be useful. The predominant variants during the study period were delta and omicron (BA.1) [29], where sotrovimab has demonstrated activity [12,13]. However, sotrovimab have shown decreased in vitro activity against BA.2 and the subsequent variants [30,31,32]. Nevertheless, mAbs activity against contemporary SARS-CoV-2 lineages is inferred exclusively from in vitro data. In vitro neutralization assays can capture just one component of the mAbs activity, and sotrovimab have Fc effector functions [14,15]. Indeed, some authors have proposed that sotrovimab has maintained in vivo efficacy against newer variants [33]. One recent study that examined mAbs activity in post-infusion sera from treated patients showed that BA.2 was neutralized by sotrovimab [34]. Moreover, recent studies have shown clinical efficacy in high-risk patients with mild COVID-19 [35,36]. Multi-omics profiles have shown the importance of immune-cell alterations in developing severe COVID-19 [37,38]. For example, monocytes show phenotypes with a decreased antigen presentation and an increased inflammatory potential [39], and elevated KIR+ CD8+ T lymphocytes have been found in patients with severe COVID-19 [40]. Sotrovimab could have the potential benefit of improving these phenotypes changes via its Fc effector functions [14]. Therefore, sotrovimab could also be a therapeutic alternative for hospitalized patients with severe COVID-19, even in a setting of new, mutant-rich variants with an apparent in vitro lack of inhibition.

Our study is a retrospective uncontrolled cohort study and, as such, has inherent limitations. The main limitations are the sample size and the absence of a control group, which limited the statistical analyses and prevented us from extracting robust conclusions. However, our rate of respiratory progression appears to be lower than what is described in recent cohorts, including breakthrough COVID-19 [22,25]. Secondly, as we had a low rate of respiratory progression, we could not perform a multivariate analysis to adjust for confounding factors when comparing the efficacy of sotrovimab in different subgroups of patients. However, our results are biologically plausible, as those patients with fewer days of symptoms and those with less severity could benefit from an agent with antiviral activity. Another limitation is that some of our patients could have been infected by the omicron variant, which may entail a lesser severity. However, we included patients with symptom onset up to December 2021, when the predominant variant was delta [23]. Finally, we could not analyze the effect of sotrovimab on immune cell functions [37,38] or on the post-acute COVID-19 syndrome [27]. Further and controlled studies are needed to clarify the potential benefit of sotrovimab in immunocompromised patients with severe COVID-19. However, our study may pose as the direction for future research.

## 5. Conclusions

In conclusion, mAbs treatment with sotrovimab appears to be safe in immunocompromised patients who have been hospitalized for severe COVID-19. Patients with fewer than 15 days of symptom duration and low-oxygen requirements may benefit from this treatment. Thus, our study poses the basis for future research to corroborate the efficacy and safety of sotrovimab in vaccinated immunosuppressed patients who have been hospitalized with severe COVID-19. Further and larger studies are needed to confirm the efficacy of this treatment for this subset of patients.

## Figures and Tables

**Figure 1 jcm-12-00864-f001:**
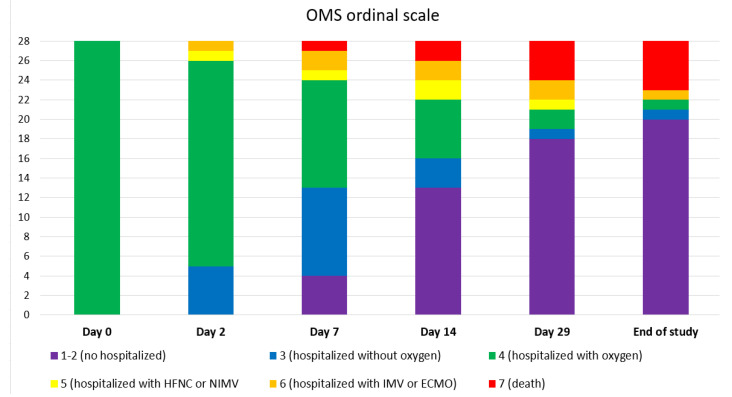
Clinical status according to OMS ordinal scale of COVID-19 severity at different days after sotrovimab infusion. Day 0 equals the day of sotrovimab infusion.

**Figure 2 jcm-12-00864-f002:**
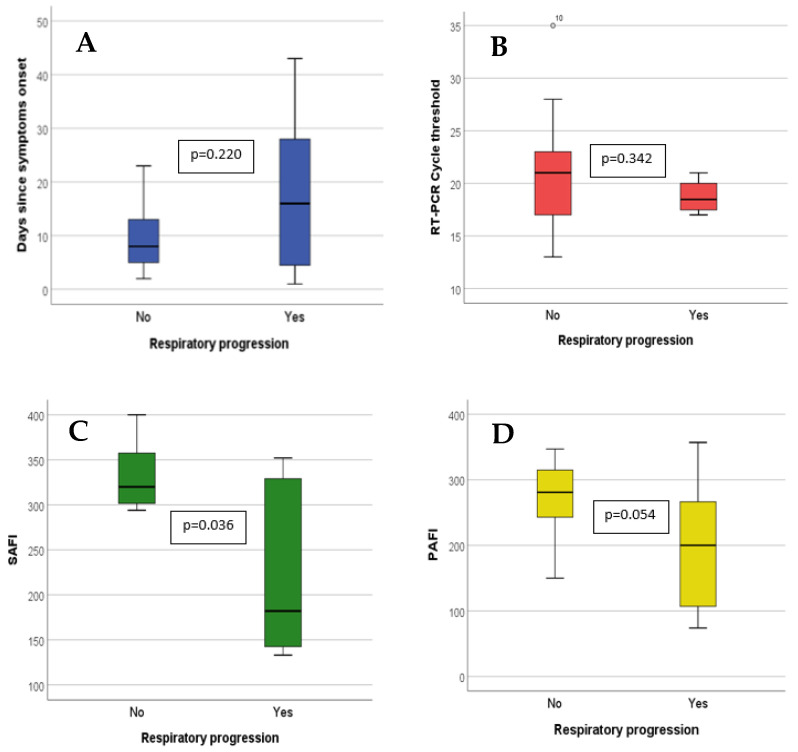
Quantitative variables value at time of sotrovimab infusion in patients with and without respiratory progression. The bar limits represent the interquartile range (p25 and p75). The black line within the bar represents the median. The lines that are external to the bar represent range (minimum–maximum). Comparison was made by means of Mann–Whitney’s U. **A:** Days since symptoms onset; **B**: RT-PCR cycle threshold; **C:** SaFi; and **D:** PaFi.

**Table 1 jcm-12-00864-t001:** Patients’ characteristics among those with and without respiratory progression.

Variable	Total (*n* = 32)	Respiratory Progression (*n* = 7)	No Respiratory Progression (*n* = 25)	*p*
Demographic and comorbidities				
Age (years)	67 (59–76)	62 (59–75)	68 (59–76)	0.721
Sex (female)	43.8% (14)	71.4% (5)	36.0% (9)	0.195
Age-adjusted Charlson Index	5 (3–6)	3 (3–6)	5 (4–7)	0.324
Obesity	21.9% (7)	28.6% (2)	20.0% (5)	1.000
Arterial hypertension	53.1% (17)	28.6% (2)	60.0% (15)	0.209
Diabetes mellitus	34.4% (11)	14.3% (1)	40.0% (10)	0.374
Chronic pneumopathy	40.6% (13)	28.6% (2)	44.0% (11)	0.671
Chronic heart failure	31.3% (10)	14.3% (1)	36.0% (9)	0.387
Liver cirrhosis	6.3% (2)	0	8.0% (2)	1.000
Chronic renal failure	37.5% (12)	14.3% (1)	44.0% (11)	0.212
Stroke	3.1% (1)	14.3% (1)	0	0.219
Chronic immunocompromise condition and medication				
Active cancer	9.4% (3)	14.3% (1)	8.0% (1)	1.000
Hematological malignancy	37.5% (12)	85.7% (6)	24.0% (6)	0.006
Solid organ transplantation	46.9% (15)	14.3% (1)	56.0% (14)	0.088
Autoimmune disease	32.3% (10)	42.9% (3)	29.2% (7)	0.652
Chronic corticoids	43.8% (14)	42.9% (3)	44.0% (11)	1.000
Calcineurin inhibitors	50.0% (16)	14.3% (1)	56.0% (14)	0.088
Antimetabolite	37.5% (12)	14.3% (1)	44.0% (11)	0.212
mTOR	6.3% (2)	0	8.0% (2)	1.000
Monoclonal antibody	37.5% (12)	71.4% (5)	28.0% (7)	0.073
Anti-CD20 antibody	28.1% (9)	57.1% (4)	20.0% (5)	0.149
Immunization status				
Fully vaccinated	93.8% (30)	85.7% (6)	96.0% (24)	0.395
Booster-dose	53.3% (16/30)	66.7% (4/6)	50.0% (12/24)	0.215
Last dose (weeks)	12 (9–29)	14 (9–26)	11 (8–30)	0.823
Serum positive anti-S IgG	9.0% (2/22)	0 (0/4)	11.1% (2/18)	1.000
COVID-19 characteristics at sotrovimab infusion				
Symptoms onset (days)	9 (5–15)	16 (3–29)	9 (5–13)	0.220
More than 14 days	22.6% (7)	57.1% (4)	12.5% (3)	0.029
Dyspnea	81.3% (26)	85.7% (6)	80.0% (20)	1.000
Cough	81.3% (26)	71.4% (5)	84.0% (21)	0.590
Fever	59.4% (19)	57.1% (4)	60.0% (15)	1.000
PaFi	273 (236–315)	200 (83–280)	287 (243–315)	0.054
PaFi greater than 210	81.3% (26)	42.9% (3)	92.0% (23)	0.005
SaFi	321 (297–351)	182 (135–336)	324 (300–354)	0.036
SaFi greater than 250	84.4% (27)	42.9% (3)	96.0% (24)	0.004
RT-PCR Ct	21 (17–23)	19 (17–20)	21 (17–23)	0.342
Other COVID-19 treatments received during admission				
Corticoids	96.9% (31)	100% (7)	96.0% (24)	1.000
Pulse-dose corticoid	3.1% (1)	0	4.0% (1)	1.000
Remdesivir	50.0% (16)	42.9% (3)	52.0% (13)	1.000
Tocilizumab	31.3% (10)	57.1% (4)	24.0% (6)	0.165
Other treatments	9.4% (3)	0	12.0% (3)	1.000

Categorical variables are expressed as percentage (absolute values) and compared by means of Chi square test (or Fisher exact test when needed). Quantitative variables are expressed as median (interquartile range) and compared by means of Mann–Whitney’s U.

## Data Availability

The datasets generated and/or analyzed during the current study are available from the corresponding author on reasonable request.

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
