# Peer review of "Use of Monoclonal Antibodies in Immunocompromised Patients Hospitalized with Severe COVID-19: A Retrospective Multicenter Cohort"

_jcm, 2023, doi:10.3390/jcm12030864_

Round 1
Reviewer 1 Report
1. What is your basis for selecting Sotrovimab for the study. Many such drugs and their analogues have already been published. Please explain it logically in Introduction.
2. The study lacks the similar results, already reported in the other papers. It seems to be a confirmatory paper for the drug, although it has already been a reported mAB. So rationalize the methods, and discussions more effectively.
Author Response
We thank the reviewer for his/hers suggestions, which have surely improved the manuscript quality. Please, find attach the point-by-point response.
- What is your basis for selecting Sotrovimab for the study. Many such drugs and their analogues have already been published. Please explain it logically in Introduction.
Response: We thank the reviewer for his/hers observation. The main reason we selected sotrovimab was that it was the only available monoclonal antibody in our country during the study period. Additionally, sotrovimab had unique qualities that made it the most suitable option: it is a “broad spectrum“ antibody that target a relatively conerved epitope in coronaviruses, it has been engineered for not producing and inflammatory response (which could be deletereus in severe COVID-19), and has the known effector function, which gives sotrovimab additional activity beyond direct viral inhibition. For these reasons, we believe that sotrovimab in concrete is a good option for these patients. We have explained these points in the introduction as requested.
- The study lacks the similar results, already reported in the other papers. It seems to be a confirmatory paper for the drug, although it has already been a reported mAB. So rationalize the methods, and discussions more effectively.
Response: The authors are not sure of what the reviewer meant in this point. Althought sotrovimab (and other mAB) have shown efficacy in preventing hospitalization in SARS-CoV2 infected patients with mild-moderate infection, there is no clear data on the efficacy and safety of these treatment in patients with severe COVID-19. To our knowledge, only a very recent published paper (Somersan-Karakaya et al, January 2023) analyzes the efficacy of a mAB in these patients. However, these paper is not focused in immunocompromised patients, as our work. We believe that the difference in the clinical profile of patients justifies the importance of the report and the new information to the literature. We acknowledge that our number of patients included is low and the lack of control group, which limits our conclusion. However, we strongly believe that our paper adds valuable information and pose hypotheses for future studies. We have added the recent publication to reference and modified the methods and discussion sections to clarify our points.
Reviewer 2 Report
Jorge Calderón-Parra et al aimed to describe the safety and efficacy of sotrovimab in severe COVID-19 in immunocompromised hosts by analyzing 32 patients immunocompromised hospitalized patients with severe COVID-19 treated with sotrovimab. They claim sotrovimab was safe and may be effective in immunocompromised patients with severe COVID-19. Although this study uniquely focused on immunocompromised patients hospitalized for server COVID-19, yet a major revision is still needed before acceptance, as the reviewer still has the following points of concern that need to be addressed.
1. This is probably my biggest concern: claiming efficacy based on only 32 patients with so much variability of immunocompromised conditions and medications is not convincing. Also, there is no clear information on where is the proper control-patient cohorts who are equally immunocompromised but not receiving sotrovimab. For this paper, at least in its current format, this reviewer is not convinced about the claims related to efficacy. The authors should either provide more data and analysis to justify that or should turn down the tone of those claims about efficacy.
2. Lack of information on how to account for other confounding factors. The authors claimed “Those receiving sotrovimab within the first 14 days from symptom onset had a lower progression risk: 12.0% (n=3/25) vs 57.1% (n=4/7), p=0.029 (figure 2).” It was not clear how did the authors statistically account for other confounding factors (e.g. gender, age, BMI etc., which may also be associated with progression risk)? If haven’t, the authors should account for these confounding factors and re-evaluate if any of these findings are still significant.
3. Safety evaluation. No quantitative statistical comparison to evaluate the safety of sotrovimab-treated versus un-treated in immunocompromised patients. Such analysis needs to be performed.
4. Molecular analysis of the immunological impact of sotrovimab is lacking. If possible, it will be nice to perform deep multiomic profiling on some of the patients treated with sotrovimab. Such approaches has already been utilized in the context of COVID-19 to investigate the molecular underpinnings of covid-19 severity (e.g. PBMID: 33171100, 33969320) and mortality (e.g. PBMID: 34489601). This will be a good opportunity to deepen the current analysis. However, the reviewer understands that it may not be easy to obtain some patient specimens. If that’s the case, this multi-omic angle should be included at least in the discussion section as a future direction, with the relevant literature cited and discussed.
5. Long covid connections. Long covid is becoming a major global health problem. Immunocompromised patients can be at risk of long covid. What treatment can be beneficial to prevent long covid in this cohort will also be of great interest. What fractions of those patients getting the sotrovimab suffer from long covid patients later? It has been found that antiviral treatment can reduce the risk of long covid in the non-immunocompromised cohort (e.g. PBMID: 35216672). However, such analysis has but not yet been performed for sotrovimab-treated immunocompromised patients. That will be a good opportunity to deepen the current manuscript. However, the reviewer understands that it may not be easy to obtain some patients' follow-up long covid information. If that’s the case, this angle should be included at least in the discussion section with the relevant literature discussed.
Author Response
We thank the reviewer for his/hers suggestions, which have surely improved the manuscript quality. Please, find attached the point-by-point response.
Jorge Calderón-Parra et al aimed to describe the safety and efficacy of sotrovimab in severe COVID-19 in immunocompromised hosts by analyzing 32 patients immunocompromised hospitalized patients with severe COVID-19 treated with sotrovimab. They claim sotrovimab was safe and may be effective in immunocompromised patients with severe COVID-19. Although this study uniquely focused on immunocompromised patients hospitalized for severe COVID-19, yet a major revision is still needed before acceptance, as the reviewer still has the following points of concern that need to be addressed.
- This is probably my biggest concern: claiming efficacy based on only 32 patients with so much variability of immunocompromised conditions and medications is not convincing. Also, there is no clear information on where is the proper control-patient cohorts who are equally immunocompromised but not receiving sotrovimab. For this paper, at least in its current format, this reviewer is not convinced about the claims related to efficacy. The authors should either provide more data and analysis to justify that or should turn down the tone of those claims about efficacy.
Response: We thank the reviewer for this commentary. It should be clarify that our work is a cohort description of highly immunocompromised patients with severe COVID-19 treated sotrovimab. We do not have available a direct control group, as we acknowledge as the main limitation of the study, together with the low sample number. We could only indirectly compare our results with what is previous cohorts of patients similarly immunosuppressed (see discussion section). Accordingly, we have tried not to transmit robust conclusions about efficacy, as we are aware of the limitations of our work. We have now modified the discussion and conclusion section to lower even more our tone about efficacy of the treatment. That been said, we strongly believe that our work provide hypothesis and direction for future research, by focusing the use of monoclonal antibodies, such as sotrovimab, in immunocompromised patients, especially those who lack adequate humoral response after vaccination.
- Lack of information on how to account for other confounding factors. The authors claimed “Those receiving sotrovimab within the first 14 days from symptom onset had a lower progression risk: 12.0% (n=3/25) vs 57.1% (n=4/7), p=0.029 (figure 2).” It was not clear how did the authors statistically account for other confounding factors (e.g. gender, age, BMI etc., which may also be associated with progression risk)? If haven’t, the authors should account for these confounding factors and re-evaluate if any of these findings are still significant.
Response: We agree with the reviewer that the significance of our conclusions in subgroup are not robust as we could not account for confounding factors. Taking into account the low rate of events (only 7 progressions), a multivariable analyses was not possible. We limited our statistical analyses to a univariate comparison. However, our results have biological plausibility. Those patients with less symptoms duration (in which virus is probably replicating actively) and those with less critical stage (in which the pulmonary damage is already done) could be the patients that benefit the most. As in the previous point, we have tried to be cautious in our conclusion about the relative efficacy in subgroups of patients. We have modified our subgroup analysis to highlight this limitation. However, our work is important as it could guide future research direction.
- Safety evaluation. No quantitative statistical comparison to evaluate the safety of sotrovimab-treated versus un-treated in immunocompromised patients. Such analysis needs to be performed.
Response: As commented above, we could not include a control group of un-treated immunocompromised patients, as our study is a cohort description, not comparison. Accordingly, we have described the adverse events found in our patients. As commented in results section, no complication or drug-related adverse event was noted during the study period, while all adverse events described were attributed to COVID-19 infection. These finding supports the safety of sotrovimab in these patients, although, as commented above, more studies are needed before extracting robust conclusions. Once again, we strongly believe in the importance of our report, as it could guide future research to study the safety and efficacy of this treatment in immunocompromised patients.
- Molecular analysis of the immunological impact of sotrovimab is lacking. If possible, it will be nice to perform deep multiomic profiling on some of the patients treated with sotrovimab. Such approaches has already been utilized in the context of COVID-19 to investigate the molecular underpinnings of covid-19 severity (e.g. PBMID:33171100, 33969320) and mortality (e.g. PBMID: 34489601). This will be a good opportunity to deepen the current analysis. However, the reviewer understands that it may not be easy to obtain some patient specimens. If that’s the case, this multi-omic angle should be included at least in the discussion section as a future direction, with the relevant literature cited and discussed.
Response: We thank the reviewer for the suggestion. We agree that sotrovimab could impact in the immunological response to COVID-19 (mainly by the effector functions mentioned in the discussion), with potential benefit in the patient’s evolution beyond antiviral in-vitro activity. A deep multiomic analysis would be of great interest in this regard. However, unfortunately, we could not retrieve patients specimens and, additionally, we do not have availability to perform this kind of studies (we do not have the technical equipment of the economic support to perform it). As suggested, we have include this as a future direction and commented in the discussion section, adding the relevant literature.
- Long covid connections. Long covid is becoming a major global health problem. Immunocompromised patients can be at risk of long covid. What treatment can be beneficial to prevent long covid in this cohort will also be of great interest. What fractions of those patients getting the sotrovimab suffer from long covid patients later? It has been found that antiviral treatment can reduce the risk of long covid in the non-immunocompromised cohort (e.g. PBMID:35216672). However, such analysis has but not yet been performed for sotrovimab-treated immunocompromised patients. That will be a good opportunity to deepen the current manuscript. However, the reviewer understands that it may not be easy to obtain some patients' follow-up long covid information. If that’s the case, this angle should be included at least in the discussion section with the relevant literature discussed.
Response. As in the previous point, we thank the reviewer for his/hers suggestion. We agree that post-acute COVID syndrome is a major health issue as it can affect a significant proportion of COVID infected patients. However, we could not perform a long-term follow up in our patients and we have not available data on the effect of sotrovimab in post-acute COVID syndrome. As suggested, we have added this to the discussion to point future research direction.
Round 2
Reviewer 1 Report
I am sorry but still the paper has several typographical errors. See carefully.
Author Response
We thank the reviewer for their comments. Please, find attached the point-by-point response.
I am sorry but still the paper has several typographical errors. See carefully.
Response: We thank the reviewer for the comment. have review the manuscript and corrected typographical errors.
Reviewer 2 Report
The reviewer still has some additional question,
1. did the authors observed similarly parallized monocytes phenotype with decreased antigen presentation but increase inflammatory potential as described in a non-immunocompromised cohort in PMID: 33171100. If so, will the sotrovimab treatment reduce such compromised monocytes?
2. Also, it has been shown that certain novel KIR+ CD8+ T cells can directly eliminate pathogenic T cells in context of COVID. Do the authors observe such phenotypes in their immunocompromised cohort and its impact by sotrovimab treatment?
These novel immune cell phenotypes will be worthwhile to investigate and/or discuss within the context of literature.
Author Response
We thank the reviewer for his/her comments. Please, find attached the point-by-point response.
The reviewer still has some additional question,
- did the authors observed similarly parallized monocytes phenotype with decreased antigen presentation but increase inflammatory potential as described in a non-immunocompromised cohort in PMID: 33171100. If so, will the sotrovimab treatment reduce such compromised monocytes?
- Also, it has been shown that certain novel KIR+ CD8+ T cells can directly eliminate pathogenic T cells in context of COVID. Do the authors observe such phenotypes in their immunocompromised cohort and its impact by sotrovimab treatment?
These novel immune cell phenotypes will be worthwhile to investigate and/or discuss within the context of literature.
Response: The reviewer poses two relevant question about the mechanism of action of sotrovimab. Phenotype of monocytes and lymphocytes T has been shown important in the response to SARS-CoV2 infection and sotrovimab, via its Fc effector functions, could potentially improve antigen presentation of monocytes, and increase CD8+ T cell action (see reference 14th of the manuscript). However, the study of the phenotype response of immune cells and its importance on clinical response is not an objective of our study, which was mainly a clinical description of patients treated with this monoclonal antibody. We are not able to perform a phenotypical analyses, as we do not have the necessary equipment or financial support to perform it. Nevertheless, we agree with the reviewer that these immune cell phenotypes are worthwhile to discuss, and we have added relevant literature to the discussion. We thank the reviewer for his/her suggestions.